# Three years of insecticide resistance evolution and associated mechanisms in *Aedes aegypti* populations of Ouagadougou, Burkina Faso

**Félix Yaméogo[1], Aboubacar Sombié[1], Manabu Oté[2,3], Erisha Saiki[2,4], Tatsuya Sakurai[2,4], Dimitri W. Wangrawa[1,5], Philip J. McCall[6], David Weetman[6], Hirotaka Kanuka[2,3], Athanase Badolo[1] ***

**1** Laboratoire d'Entomologie Fondamentale et Appliquée, Université Joseph Ki-Zerbo, Ouagadougou, Burkina Faso, **2** Center for Medical Entomology, The Jikei University School of Medicine, Tokyo, Japan, **3** Department of Tropical Medicine, The Jikei University School of Medicine, Tokyo, Japan, **4** Laboratory Animal Facilities, The Jikei University School of Medicine, Tokyo, Japan, **5** Université Norbert Zongo, Koudougou, Burkina Faso, **6** Department of Vector Biology, Liverpool School of Tropical Medicine, Liverpool, United Kingdom

* a.badolo@gmail.com

**Data Availability Statement:** All data generated and/or analyzed during this study are included in

## Abstract

### Background

Resistance to insecticides is spreading among populations of *Aedes aegypti*, the primary vector of important human arboviruses. The escalating insecticide resistance poses a significant threat to dengue vector control, with an expanding number of countries affected by the disease. To gain a deeper insight into the evolution of insecticide resistance, it is essential to have longitudinal surveillance results, which are currently lacking, particularly from African *Ae. aegypti* populations. Here we report on three-years of surveillance of *Ae. aegypti* susceptibility to insecticide resistance phenotypes and associated *kdr* mutations in Burkina Faso, a country with regular dengue outbreaks.

### Methods

*Ae. aegypti* susceptibility to insecticides and the V410L, V1016I, and F1534C *kdr* target site mutations linked to pyrethroid insecticide resistance were monitored in Ouagadougou from 2016 to 2018. Larvae were collected from artificial containers at two sites and reared to adulthood in an insectary. Bioassays were conducted on female adults, along with a laboratory-susceptible strain, following standard WHO protocols. Allele-specific PCR genotyping assays were utilized to identify the V410L, V1016I, and F1534C *kdr* pyrethroid target site mutations.

### Results

Bioassays revealed a high level of resistance to permethrin and deltamethrin that progressively increased over the three-year period in both localities. The 1534C mutation was nearly fixed throughout the three years at each locality, and while the closely-related 410L and 1016I mutations did not vary between localities, their frequency notably increased from

the manuscript of provided as supplementary information.

**Funding:** This work was supported by a WHO/TDR grant (RCS-KM 2015 ID235974) to AB, DW and PJM, and the International Collaborative Research Program for Tackling the NTDs Challenges in African countries from Japan Agency for Medical Research and Development, AMED (JP17jm0510002h0003) to HK and AB. PJM's research on peri-domestic behavior of Aedes aegypti receives support from MRC-UK (MR/T001267/1). The funders had no role in the study design, data collection and analysis, decision to publish, or preparation of the manuscript.

**Competing interests:** The authors have declared that no competing interests exist.

2016 to 2018. Interestingly, *Ae. aegypti* populations in both areas remained susceptible to bendiocarb, fenitrothion, and malathion. Modelling the mortality data further confirmed the escalating resistance trend over the years and emphasized the significant role played by the three kdr mutations in conferring resistance to pyrethroids.

## Conclusion

Mortality rates indicate that *Ae. aegypti* populations from Ouagadougou are becoming increasingly resistant to pyrethroid insecticides, likely due to an increase in the frequencies of the 410L and 1016I *kdr* mutations. Organophosphate insecticides are likely to be better alternative options for control.

## Author summary

*Aedes aegypti* transmits diseases including dengue, chikungunya, and Zika. This vector has developed resistance to several insecticides used in vector control interventions. Resistance data for *Ae. aegypti* in Africa remain sparse, and the evolution of insecticide resistance is still under-investigated. We undertook three-years of monitoring of insecticide resistance in *Ae. aegypti* and the associated *kdr* mutations in two localities of Burkina Faso. Larvae were collected from two different breeding sites, tyres and drums, from both localities. Insecticide bioassay susceptibility tests to commonly used insecticides were performed in the laboratory. The results revealed escalating resistance to pyrethroid insecticides, supported by increasing frequencies of the 1016I and 410L *kdr* mutations, while the 1534C mutation is fixed. Interestingly, *Ae. aegypti* populations in both areas remained susceptible to bendiocarb, fenitrothion, and malathion, providing a conservative option of organophosphates for *Ae. aegypti* control in Burkina Faso. These data provide important results to support decisions on dengue vector control.

## Introduction

Burkina Faso has a long history of dengue fever, with the first cases recorded in 1925 [1]. In 2015, several cases of dengue haemorrhagic fever were recorded in Ouagadougou [2], followed by a significant dengue outbreak in 2016, during which 1,256 suspected cases and 15 deaths were reported in the fourth quarter of the year [3]. In November 2017, a total of 9,029 cases and 18 deaths were reported throughout the country [4]

Dengue control and prevention relies primarily on vector control using larval source management [5] and chemical control of adult *Aedes* mosquitoes. This includes the use of organophosphates and pyrethroid insecticides for space spraying, and less commonly, pyrethroid-treated materials for personal or household protection [6]. In Burkina Faso, dengue outbreak control has been implemented since the 2016 and comprises of larval source management and insecticide spraying for vector control in dengue hotspots. Indoor and outdoor spraying is practiced in dengue patient houses and contiguous houses, and spatial spraying is used in public places and large larval sites.

Resistance to the four classes of insecticides most frequently used for vector control has been reported in *Ae. aegypti* populations worldwide, with particularly high prevalence in Latin

American populations, showing broad-scale regional variation in at least some of the major resistance mechanisms involved [7].

*Aedes aegypti* has rarely been the target of focused vector control in Africa, except during periods of dengue outbreaks. Nonetheless, populations are often documented to exhibit resistance to pyrethroid insecticides, especially in west and central Africa [8,9].

Pyrethroid resistance in Africa is typically associated with knockdown resistance (*kdr*) mutations in the voltage-gated sodium channel (Vgsc), especially 1534C, 1016I, and 410L [9,10].

The F1534C variant is the most widespread mutation in *Ae. aegypti*, conferring resistance to deltamethrin and permethrin often in association with one or more of over 10 other mutations, such as V1016I and V410L [7,11,12].

The V1016I mutation confers resistance to pyrethroid insecticides only when associated with the F1534C mutation in *Ae. aegypti* [11,13], whereas V410L can confer resistance alone but is mostly found in linkage with V1016I [14,15,16]. In Burkina Faso, insecticide resistance of *Ae. aegypti* has been reported since 2016 with documented resistance to pyrethroids, moderate resistance to bendiocarb and susceptibility to malathion [8]. The mechanisms involved in pyrethroid resistance in Burkina Faso primarily involve the V1016I, 1534C and 410L *kdr* mutations [8,10,17], and to a lesser extent metabolic resistance [8,10,11]. The co-evolution of the V410L mutation with V1016I and F1534C can provide elevated levels of resistance to pyrethroids [16, 18]. After several years of resistance monitoring in Brazilian *Ae. aegypti*, Macoris *et al.* [18] and Garcia *et al.* [19], documented a geographical expansion and increase in the frequency of the 1534C and 1016I mutations, as well as evidence for increasing activity of the major classes of detoxifying enzymes.

Many reports on *Ae. aegypti* resistance to pyrethroid insecticides originate from Asia and the Americas; data from Africa remain sparse and longitudinal surveys are also lacking [20]. In this study, we report on the evolution of *Ae. aegypti* susceptibility to insecticides and the frequency of *kdr* mutations over three consecutive years in two localities during the same period of the year, following the 2016 dengue outbreak in Burkina Faso.

## Methods

### Ethics statement

The research protocol entitled (16–030) "Dengue in Burkina Faso: establishing a vector biology evidence base for risk assessment and vector control strategies for an emerging disease" (16–030) received ethical approval from the Ethical Committee for health Research, Ministry of Health (Deliberation N˚2016-6-073) on 6th June 2016 and the Research Ethics Committee at the Liverpool School of Tropical Medicine on 15th July 2016.When larvae were collected inside or near a residence, permission from the owners/residents was obtained before entering their property or land and informed consent was signed. The study involves only *Aedes aegypti*, no human subject was involved.

### Study site, larval collection, and laboratory processing

Larvae and pupae of *Ae. aegypti* were collected from tires and drums during the rainy season for three consecutive years (2016–2018) from August to September. Two selected localities classified as peri-urban (Tabtenga: N12˚22′05.7″ and W001˚27′35.0″) and urban (1200 Logements N12˚22′31.4″ and W001˚29′51.3″), where dengue cases were previously reported, were chosen for mosquito sampling. The collected larvae and pupae were transferred to the insectary and reared to the adult stage using dried fish food (Tetramin) until the adult stage. Emerged adult mosquitoes were fed a 10% sugar solution for 3–5 days, and females were used

in bioassays. The insectary conditions were maintained at a temperature of 27±2˚C, a relative humidity of 77±6%, and a 12: 12 light: dark photoperiod.

## Adult susceptibility bioassays

Insecticidal bioassays were conducted following the WHO tube test protocol [21]. Five insecticides were tested: permethrin (0.75%), deltamethrin (0.05%), bendiocarb (0.1%), fenitrothion (1%), and malathion (5%). The discriminating doses of fenitrothion and bendiocarb correspond to those prescribed for *Ae. aegypti*, whereas those for permethrin, deltamethrin, and malathion recommended for *Anopheles gambiae*, and though very commonly used [7], are higher than those for *Ae. aegypti*. Approximately 100 3–5-day-old unfed females were exposed to insecticide-impregnated papers (ordered from the Universiti Sains Malaysia WHO reference centre), and the number knocked down was counted every 10 minutes for one hour. Two tubes with non-impregnated paper, each containing 25 mosquitoes, were used as controls. After the exposure period, mosquitoes were transferred to holding tubes for 24 hours, maintaining the same temperature and humidity conditions as above. The fully insecticide-susceptible Rockefeller strain of *Ae. aegypti* was used as a negative control to ensure the validity of the tests (100% mortality is expected for this strain). After 24 hours of observation, dead mosquitoes were counted, and the mortality rates were adjusted according to Abbott's formula [22] when the control tube's mortality rate ranged between 5 and 20%. Dead and alive mosquitoes were preserved in 1.5 ml micro-tubes over silica gel and stored at -20˚C for molecular work.

Susceptibility tests were also performed following the CDC bottle bioassay protocol [17] to investigate potential metabolic resistance involvement in pyrethroid resistance through synergist effects, but interpretation was made according to WHO guidelines [23]. Bottle bioassays were used for synergist tests for all years because PBO-impregnated papers were unavailable for the work in 2016. Permethrin and deltamethrin were used to coat bottles using acetone as a solvent for final concentrations of 0.015% (15μg) and 0.01% (10μg), respectively, as recommended for *Ae. aegypti*. For each bioassay using the synergist, approximately 150 non-blood-fed mosquitoes of 3–5 days were exposed first to the synergist (synergist exposure bottles) with 25 mosquitoes per bottle for 1 hour. After exposure to PBO, 100 mosquitoes were transferred to pyrethroid-coated bottles and 50 mosquitoes to acetone-coated bottles as insecticide control for 1 hour [17]. Two bottles were coated with acetone only as synergist controls. The insecticides and synergist PBO used were ordered from Sigma-Aldrich at technical grade purity (>98% purity). At the end of each exposure period, mosquitoes were removed and kept in cardboard cups for 24 hours of observation. At the end of the 24-hour observation time, dead and alive mosquitoes were preserved in 1.5 ml tubes over silica gel and stored at -20˚C. Bioassay data from 2016 were published in [8]but are included for the purpose of comparison.

## DNA extraction and genotyping of V410L, V1016I and F1534C *kdr* mutations

DNA was extracted from mosquitoes using an ethanol precipitation method [24]. Genotyping of the F1534C *kdr* mutation was performed following the protocol of Li et *al.* [25] in a two-step PCR method with two common primers and one specific primer for each allele. Genotypes of the 1016I mutation were detected using an allele specific PCR according to the protocols of Saavedra-Rodriguez et *al.* [14] and Martins et *al.* [26]. Genotyping of the V410L *kdr* mutation followed the protocol described by Granada et *al.* [13], with the wild-type and mutant alleles identified in two separate PCR reactions.

For all PCRs conducted, each reaction tube contained 1 μl of target DNA, 6.25 μl of enzyme (AmpliTaq, Thermo Fisher Scientific), and 0.3 μM of each primer for a total volume adjusted

to 12.5 μl with distilled water. Subsequently, 5μl of the PCR product was mixed with 6x loading buffer (New England BioLabs) and electrophoresed on a 1.5% TAE agarose gel for F1534C and V410L, and 3% TBE for V1016I. The gels were then stained with ethidium bromide and visualized under ultraviolet (UV) light. A total of 286 *Ae. aegypti* samples were genotyped over three years: 94 samples in 2016, 98 samples in 2017 and 94 samples in 2018. Control (untested) specimens were tested in 2016 and pyrethroid-exposed specimens in 2017 and 2018.

### Statistical analysis

Mortality was calculated as the percentage of individuals that died within 24 hours after 1 hour exposure to insecticide. Bioassay outcomes were interpreted according to the WHO protocol [21]. Mosquitoes were considered resistant to an insecticide when mortality was less than 90%, and susceptible when mortality ranged between 98–100%. Resistance to insecticide is suspected when mortality is between 90–97% and requires subsequent testing for confirmation. The 95% confidence limits of mortality were calculated based on the method of Newcombe [27]. A generalized linear model (glm) with a binomial link function was fitted to the data to assess the effects of insecticide-locality-year and insecticide-locality, locality-year interactions on mosquito mortality.

Moreover, the relationship between genotypes or haplotypes and phenotype (dead/alive) of mosquitoes was determined using a glm model using a binomial function. All analyses were performed using R software version 4.3.1. Significance level throughout was set at $p < 0.05$.

## Results

### Adult susceptibility bioassay

The Rockefeller strain exhibited full susceptibility to all tested insecticides, leading to 100% mortality in WHO bioassays.

*Ae. aegypti* populations from 1200 Logements and Tabtenga remained fully susceptible to malathion and fenitrothion (100% mortality), during the three years of monitoring (Fig 1A and 1B). The glm showed a significant difference ($p < 0.001$) in mortality for fenitrothion in comparison with bendiocarb as reference, but not for malathion (p = 0.98; Table 1).

Bendiocarb resistance was observed in both 1200LG and Tabtenga in 2016 (mortality<90%), but the status changed to suspected resistance in both localities in 2017 (mortality 90–97%), with a further reclassification in 2018 to susceptibility (mortality >98%; Fig 1A and 1B).

WHO tubes bioassays revealed a high prevalence of resistance to permethrin and deltamethrin in each population across the three years with significantly decreasing mortality to both pyrethroids between 2016 and 2018 ($p < 0.001$; Tables 1 and S1 and Fig 1A and 1B). Additionally, the glm showed a lesser effect of pyrethroid insecticide on mortality as indicated by odds ratios values compared to the reference bendiocarb (Table 1).

The trend of insecticide resistance using CDC bottles bioassays at recommended concentrations for *Ae. aegypti* mirrored WHO tube results for deltamethrin and permethrin, with generally decreasing mortalities over the three years for 1200 Logements and Tabtenga (Fig 1C and 1D).

The combination of permethrin or deltamethrin with the PBO synergist partially restored the susceptibility. However, with increasing resistance, the absolute degree of restoration also decreased across years, though synergism ratios increased. The glm revealed a significant difference between mortalities in *Ae. aegypti* populations from 1200LG and Tabtenga during years 2016, 2017 and 2018, likely driven by slightly different responses to permethrin between sites across the years (Table 2).

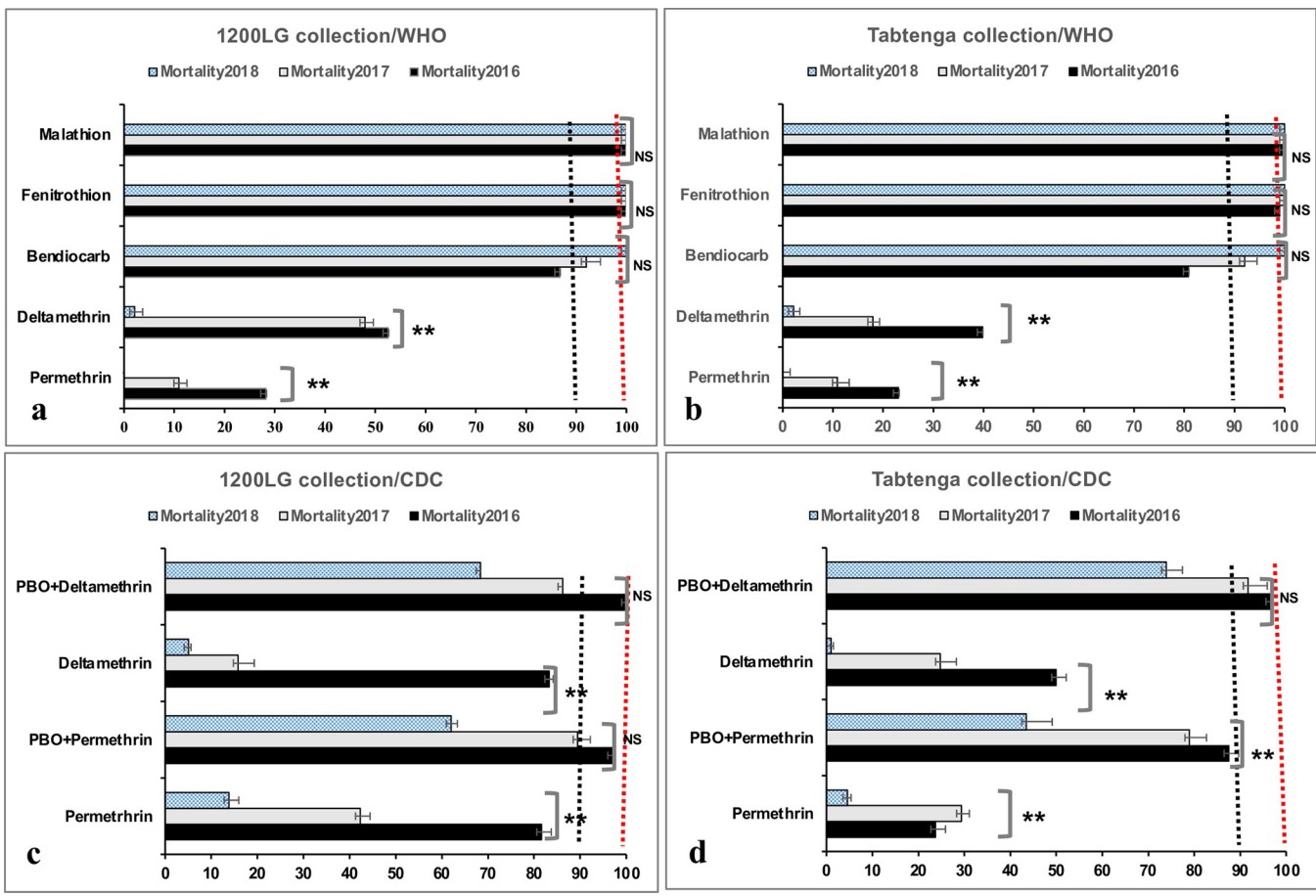

**Fig 1. Mortality of *Ae. aegypti* adults exposed to insecticides.** WHO tubes and CDC bottles with 1 hour exposure to insecticides and synergist mortality (x-axis) and the 95% confidence limits of 1200LG (1200 Logements) and Tabtenga collection. The significance of the comparison between the three years for each locality is given: (**) significant, (NS) Not significant). The red and black dashed lines indicate susceptibility and resistance thresholds, respectively.

### Evolution of F1534C, V1016I and V410L *kdr* mutations frequencies

The F1534C mutant exhibited high allelic frequency, nearing fixation in *Ae. aegypti* populations from both localities in 2016 and becoming fixed by 2018 (Fig 2A and 2B). There were no wild-type homozygotes recorded in either locality, and heterozygotes were extremely uncommon (range 0–0.03).

The frequency of the 1016I allele increased between 2016 and 2018 in both Tabtenga and 1200LG. The genotypic frequency of homozygous 1016I/I was lower in *Ae. aegypti* from 1200LG and Tabtenga in 2016, with the frequency of this genotype increasing in 2017 and 2018 in both localities (S1B Fig). The 410L allele frequency also increased from 2016 to 2018 in 1200LG (Fig 2A) and in Tabtenga during the same period (Fig 2B). The genotype frequency of homozygous 410L/L increased between 2016 and 2018 for populations of 1200LG (S1A Fig) and Tabtenga (S1D Fig). The heterozygote (410V/L) frequency also increased in both localities from 2016 to 2018, in association with the wild allele (V410) frequency decreasing from 2016 to 2018 (S1A-S1D Fig).

### Evolution of F1534C, V1016I and V410L haplotypes, genotypes and their association with pyrethroid resistance

Five out of a total of eight possible haplotypes (CIL, CIV, CVL, CVV and FVV) were recorded from the genotypes of the three variants in the two localities, during 2017 and 2018 in *Ae*.

**Table 1. Parameters of the generalized linear model of *Ae. aegypti* mortality in 2016, 2017 and 2018 using WHO test.** For each predictor, the incidence rate ratio, confidence limits and corresponding p-values are provided.

| Predictors | Odds Ratios | CI | P |
|---|---|---|---|
| (Intercept) | 16.63 | 11.42–24.90 | <0.001 |
| Insecticide [Bendiocarb] | | | |
| **Deltamethrin** | **0.04** | **0.03–0.06** | **<0.001** |
| **Fenitrothion** | **11.83** | **4.22–49.37** | **<0.001** |
| Malathion | – | Inf–Inf | – |
| **Permethrin** | **0.02** | **0.01–0.03** | **<0.001** |
| Locality [1200LG] | | | |
| Tabtenga | 0.69 | 0.41–1.17 | 0.172 |
| Year [2016] | | | |
| **2017** | **0.46** | **0.31–0.66** | **<0.001** |
| **2018** | **0.33** | **0.22–0.47** | **<0.001** |
| Insecticide [Bendiocarb] x Locality [1200LG] | | | |
| Deltamethrin x Tabtenga | 0.78 | 0.44–1.39 | 0.394 |
| Fenitrothion x Tabtenga | 2.64 | 0.41–20.94 | 0.304 |
| Malathion x Tabtenga | – | – | – |
| Permethrin x Tabtenga | 0.86 | 0.49–1.53 | 0.604 |
| Locality [1200LG] x Year [2016] | | | |
| Tabtenga x 2017 | 1.54 | 0.92–2.59 | 0.099 |
| Tabtenga x 2018 | 1.26 | 0.72–2.21 | 0.421 |

Significant predictors are highlighted in bold. Non-significant terms were interactions Insecticide*Locality, Locality*Year and Insecticide*Locality*Year. No survivors were recorded from the malathion exposure.

**Table 2. Parameters of generalized linear model of *Ae. aegypti* mortality in 2016, 2017 and 2018 using CDC bottles test.** For each predictor, the incidence rate ratio, confidence limits and corresponding p-values are provided.

| Predictors | Odds Ratios | CI | P |
|---|---|---|---|
| (Intercept) | 3.92 | 2.83–3.53 | <0.001 |
| Insecticide [Deltamethrin] | | | |
| **PBO + Deltamethrin** | **29.17** | **17.34–50.65** | **<0.001** |
| **PBO + Permethrin** | **24.74** | **14.89–42.36** | **<0.001** |
| **Permethrin** | **2.16** | **1.39–3.41** | **<0.001** |
| Locality [1200LG] | | | |
| **Tabtenga** | **0.20** | **0.13–0.31** | **<0.001** |
| Year [2016] | | | |
| **2017** | **0.07** | **0.04–0.11** | **<0.001** |
| **2018** | **0.02** | **0.01–0.03** | **<0.001** |
| Insecticide [Deltamethrin] x Locality [1200LG] | | | |
| PBO + Deltamethrin x Tabtenga | 1.44 | 0.71–2.90 | 0.315 |
| **PBO + Permethrin x Tabtenga** | **0.44** | **0.23–0.82** | **0.011** |
| **Permethrin x Tabtenga** | **0.29** | **0.16–051** | **<0.001** |
| Locality [1200LG] x Year [2016] | | | |
| **Tabtenga x 2017** | **6.67** | **3.83–11.79** | **<0.001** |
| **Tabtenga x 2018** | **4.74** | **2.44–9.34** | **<0.001** |

Significant predictors are highlighted in bold. The non-significant third-order interaction term Insecticide*Locality*Year was removed from the model.

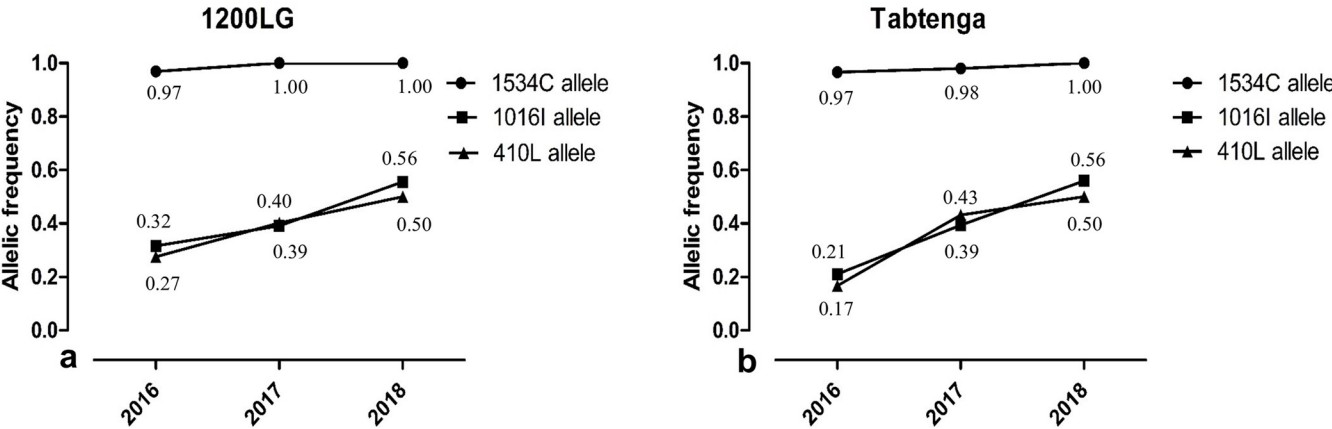

**Fig 2. Evolution of *kdr* mutant allele frequencies (1534C, 1016I and 410L) in *Ae. aegypti* collected in 1200LG and Tabtenga during 2016, 2017 and 2018.**

*aegypti* exposed to permethrin and deltamethrin (Tables 3 and S1). Data from 2016 are published [8], though without detection of the V410L *kdr* mutation at that time.

The triple mutation 1534C/1016I/410L haplotype (CIL) and the single (1534) mutant haplotype CVV were the most prevalent in *Ae. aegypti* populations of the two localities (S2 Table), while CVL was rare. Only the CIL haplotype increased the survival of the mosquito (approximately three-fold) compared to the reference CVV. The other haplotypes, CIV and FVV did not show a significant difference in survival to both permethrin and deltamethrin (p>0.05) (Tables 3 and S1). Sample size limited the assessment of contribution of most haplotypes' associations to insecticide resistance (Tables 3 and S1).

The frequency of the CIL/CIL triple mutant genotype increased from 2016 to 2018 (S2 Table) in 1200LG (0.020 to 0.20) and Tabtenga (0.04 to 0.24), whereas the CVV/CVV genotype frequency decreased in both localities (1200LG: 0.37 to 0.11; Tabtenga: 0.58 to 0.18) between 2016 and 2018 (S2 Table). The wild type genotype FVV/FVV was absent and the FVV/CVV was infrequent during the first two years, and absent in 2018. As the 1534C allele is almost

**Table 3. Generalised linear model of survival of *Ae. aegypti* to permethrin and deltamethrin from 1200LG and Tabtenga collected in 2017 and 2018. For each predictor, the incidence rate ratio, confidence limits and corresponding p-values are provided.**

| Predictors | Odds Ratios | CI | P |
|---|---|---|---|
| (Intercept) | 0.50 | 0.26–0.93 | 0.032 |
| Insecticide [Deltamethrin] | | | |
| **Permethrin** | **14.45** | **4.89–54.12** | **<0.001** |
| Locality [1200LG] | | | |
| Tabtenga | 0.54 | 0.24–1.23 | 0.146 |
| Year [2017] | | | |
| **2018** | **22.87** | **6.06–151.09** | **<0.001** |
| Haplotypes [CVV] | | | |
| **CIL** | **2.88** | **1.52–5.61** | **0.001** |
| CIV | 1.57 | 0.26–13.39 | 0.642 |
| FVV | 3.69 | 0.14–98.33 | 0.370 |
| Insecticide [Deltamethrin] × Locality [1200LG] | | | |
| Permethrin × Tabtenga | 0.32 | 0.07–1.28 | 0.121 |
| Locality [1200LG] × Year [2017] | | | |
| Tabtenga × 2018 | 2.76 | 0.29–26.58 | 0.350 |

fixed, a glm of the genotypes was based on the two mutations 1016I and 410L and showed that the IL/IL genotype increased the survival of *Aedes* mosquitoes six-fold compared to the reference genotype VV/VV (S3 Table).

## Discussion

Pyrethroid resistance in *Ae. aegypti*, reported in 2016 and 2017, was linked to a high frequency of the 1016I and 1534C *kdr* mutations [8, 24]. Our current study presents a three-year follow-up, showing an increasing insecticide resistance to pyrethroid insecticides associated with the 1534C, 1016I, and 410L *kdr* mutations in *Ae. aegypti* populations from Tabtenga and 1200LG in Ouagadougou. These populations exhibited high and increasing pyrethroid resistance over the surveillance period, with incomplete restoration of susceptibility when the PBO synergist preceded pyrethroid exposure. At the time of the study, impregnated papers at *Aedes* specific doses were not available. Most of the doses used in our bioassays were *Anopheles* doses, slightly higher than those for *Aedes*, providing a conservative assessment of *Ae. aegypti* resistance. Recently these doses have been revised by the WHO [28].

The first documented West African case of *Ae. aegypti* pyrethroid resistance was reported in Ghana, involving the F1534C *kdr* mutation and the V1016I mutations [29]. Subsequently, resistance was also observed in Cameroon [30], though at the time without detection of *kdr* mutations. Recent studies in Ghana and Cameroon have reported significant pyrethroid resistance linked to *kdr* mutations [31,32]. Because previous studies in these countries were not carried out in the same localities, spatial variation may be a confounding factor in the temporal increase of insecticide resistance and *kdr* mutation allelic frequency recorded. The current study has the advantage of being carried out in the same localities and at the same period over three consecutive years. This allows for a consistent demonstration of the escalating resistance of *Ae. aegypti* to pyrethroid insecticides and the associated *kdr* mutations over time in the two localities.

The synergist PBO partially restored susceptibility to deltamethrin and permethrin in *Ae. aegypti*, indicating the involvement of cytochrome P450 monooxygenase activity in pyrethroid resistance, as observed in mosquitoes from the same localities with overexpression of P450 candidate genes [8]. The absolute mortality rate associated with PBO+pyrethroid decreased over the three-year period, but the relative mortality attributable to PBO synergistic effect has increased. This indicates an increased synergistic effect of PBO, likely indicating heightened oxidase activity.

Resistance was maintained for the three years of surveillance, and even more, the survivorship of *Ae. aegypti* to these doses increased. The distribution of the V410L, V1016I, and F1534C genotypes has shifted over the years, showing a rise in the allelic frequency of 410L and 1016I, while the 1534C mutation has nearly reached fixation in the population. Previous studies have established a link between insecticide application and the dynamics of resistance in *Ae. aegypti* [19], particularly the development of resistance due to prolonged insecticide use in dengue vector control campaigns [33,34].

The haplotype 1534C/1016I/410L is significantly associated with permethrin and deltamethrin survivors in both the 1200LG and Tabtenga collections, indicating that the presence of this triple mutation induces pyrethroid resistance in *Ae. aegypti*. Similar findings were highlighted in Mexico, where the association between the triple mutation and pyrethroid resistance was observed [14]. Another study conducted in Mexico from 2000 to 2012 [35] reported an increase in the frequencies of the 1534C and 1016I mutations in *Ae. aegypti*, reflecting a similar trend to that observed in the current study. The 410L mutation, either alone or in combination with 1534C, confers high levels of resistance [14], making its emergence a challenge for vector control using pyrethroids [16]. The evolution of this triple mutation over a three-

year period in the current study could largely explain the intensification of resistance observed with deltamethrin and permethrin.

The status of *Ae. aegypti* populations' resistance or susceptibility to bendiocarb is not entirely clear in the current study. Resistance decreased progressively from 2016 (mortality < 90%) to full susceptibility in 2018 in the two localities. However, an intensity test of 2 hours exposure to the diagnostic dose of bendiocarb did not confirm the 2016 resistance results in Tabtenga and 1200LG [8]. Additionally, Sombié et al. [24] confirmed the susceptibility to carbamates in Somgandé, while in 2017, resistance was recorded in the same locality, indicating spatial heterogeneity in carbamate resistance.

*Aedes aegypti* was not targeted by vector control measures in Burkina Faso prior to the 2016 outbreak where malathion was used in spatial and indoor residual spray in dengue hotspots. Previously a shift from susceptibility to full resistance within a three-year period for *Anopheles gambiae* was documented in an area with the only apparent change in selection pressure being the free distribution of insecticide treated bednets [34]. Several other studies have also made the link between bednet use and development of resistance in malaria vectors [36, 37], and also in *Culex quinquefasciatus* [38]. Long-lasting insecticidal nets (LLINs) were distributed in 2010, 2013, 2014 and 2016 in Burkina Faso increasing bednet coverage to 85% in the population [39]. Yet given the differences in behaviour between these vectors and *Ae. aegypti*, particularly in Africa, selection pressure from bednets is expected to be considerably less strong for *Ae. aegypti*. Urban agricultural use of insecticides in gardening may constitute some selection pressure for *Ae. aegypti* resistance to insecticides [40]. However, insecticide used in larger-scale agriculture probably has a limited effect as resistance pressure on *Ae. aegypti*, given that it primarily breeds in artificial containers in urban settings, even if physicochemical factors that affect mosquito body size [41] may also play a role in selecting resistance to insecticides [42]. Overall, the lack of a clear selective driver supports the idea of multiple selection routes in the domestic environment potentially involving impregnated bednets, insecticide aerosols and mosquito coils [43]. Selective pressure from domestic insecticide use may be quite significant because recent studies have shown that *Ae. aegypti* exhibit primarily exophilic behaviour but with an apparent tendency to bloodfeed more indoors than expected [43] meaning that insecticide-treated bednets and other sources of pyrethroid insecticides used indoors may play some role in the development of insecticide resistance. However, uncertainty clearly remains and the drivers of insecticide resistance in *Ae. aegypti* in Burkina Faso and elsewhere in Africa require further investigation.

## Conclusion

*Ae. aegypti* maintained resistance to pyrethroid insecticides over the three years of surveillance, with resistance levels increasing over time. The F1534C, V410L, and V1016I *kdr* frequencies increased, with the F1534C *kdr* mutation reaching fixation, and the homozygotes for both V410L and V1016I surviving pyrethroid insecticide exposure. *Aedes aegypti* remained susceptible to fenitrothion and malathion at the tested doses. These findings have significant implications for vector control planning against arbovirus vectors in Burkina Faso and may provide valuable insights for other African countries with limited data on insecticide resistance. Given the escalating insecticide resistance, urgent attention is required to develop insecticide resistance-mitigating strategies to prevent arbovirus outbreaks.

## Supporting information

**S1 Fig. Evolution of genotypic frequencies of F1534C, V1016I and V410L mutations in *Ae. aegypti* mosquitoes from 1200LG and Tabtenga during 2016, 2017 and 2018.**
(PDF)

**S1 Table. Mortality and genotypes of pyrethroid exposed mosquitoes.**
(XLSX)

**S2 Table. Co-occurrence of F1534C, V1016I and V410L mutations in 286 samples of Ae. aegypti populations.** The numbers of each genotype are provided, along with frequencies in parentheses. The probability for comparison of the genotype distributions over the three years for each locality was determined using the Fisher Exact test.
(DOCX)

**S3 Table. Glm of *survivor* genotypes in *Ae. aegypti* to permethrin and deltamethrin from 1200LG and Tabtenga collected in 2017 and 2018.** For each predictor, the incidence rate ratio, confidence limits and corresponding p-values are provided.
(DOCX)

## Acknowledgments

The authors are grateful to the members and leaders of the communities in 1200 Logements and Tabtenga for permitting larval collection on their properties.

## Author Contributions

**Conceptualization:** Philip J. McCall, David Weetman, Athanase Badolo.

**Data curation:** Félix Yaméogo, David Weetman, Athanase Badolo.

**Formal analysis:** David Weetman, Athanase Badolo.

**Funding acquisition:** Philip J. McCall, David Weetman, Hirotaka Kanuka, Athanase Badolo.

**Investigation:** Félix Yaméogo, Aboubacar Sombié, Manabu Oté, Erisha Saiki, Tatsuya Sakurai, Dimitri W. Wangrawa, Hirotaka Kanuka, Athanase Badolo.

**Methodology:** Aboubacar Sombié, Manabu Oté, Erisha Saiki, Tatsuya Sakurai, Philip J. McCall, David Weetman, Hirotaka Kanuka, Athanase Badolo.

**Project administration:** Athanase Badolo.

**Resources:** Hirotaka Kanuka, Athanase Badolo.

**Supervision:** Athanase Badolo.

**Validation:** Philip J. McCall, David Weetman, Athanase Badolo.

**Visualization:** Félix Yaméogo, David Weetman, Athanase Badolo.

**Writing – original draft:** Félix Yaméogo, Athanase Badolo.

**Writing – review & editing:** Félix Yaméogo, Aboubacar Sombié, Manabu Oté, Erisha Saiki, Tatsuya Sakurai, Dimitri W. Wangrawa, Philip J. McCall, David Weetman, Hirotaka Kanuka, Athanase Badolo.

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
