## [Decision Letter · Decision Letter 0]

30 May 2024

Dear Dr. Yameogo,

Thank you very much for submitting your manuscript "Three years of insecticide resistance evolution and associated mechanisms in Aedes aegypti populations of Ouagadougou, Burkina Faso." for consideration at PLOS Neglected Tropical Diseases. As with all papers reviewed by the journal, your manuscript was reviewed by members of the editorial board and by three independent reviewers. Based on the reviews, we are likely to accept this manuscript for publication, providing that you modify the manuscript according to the review recommendations. 

In addition to reviewers comments, I also suggest to change [*] for [NS] to indicate an absence of significance in Fig1 in order to avoid misinterpretation of the data. Finally, as suggested by reviewers, I also believe that this MS deserves to be proofread by a native English speaker before being published.

Sincerely,

Jean-philippe David

Academic Editor

Nigel Beebe

Section Editor

Reviewer's Responses to Questions

**Key Review Criteria Required for Acceptance?**

**Methods**

-Are the objectives of the study clearly articulated with a clear testable hypothesis stated?

-Is the study design appropriate to address the stated objectives?

-Is the population clearly described and appropriate for the hypothesis being tested?

-Is the sample size sufficient to ensure adequate power to address the hypothesis being tested?

-Were correct statistical analysis used to support conclusions?

-Are there concerns about ethical or regulatory requirements being met?

Reviewer #1: (No Response)

Reviewer #2: NA

Reviewer #3: (No Response)

**Results**

-Does the analysis presented match the analysis plan?

-Are the results clearly and completely presented?

-Are the figures (Tables, Images) of sufficient quality for clarity?

Reviewer #1: (No Response)

Reviewer #2: NA

Reviewer #3: (No Response)

**Conclusions**

-Are the conclusions supported by the data presented?

-Are the limitations of analysis clearly described?

-Do the authors discuss how these data can be helpful to advance our understanding of the topic under study?

-Is public health relevance addressed?

Reviewer #1: (No Response)

Reviewer #2: NA

Reviewer #3: (No Response)

**Editorial and Data Presentation Modifications?**

Reviewer #1: (No Response)

Reviewer #2: NA

Reviewer #3: (No Response)

**Summary and General Comments**

Reviewer #1: The article “Three years of insecticide resistance evolution and associated mechanisms in Aedes aegypti populations of Ouagadougou, Burkina Faso” gives an interesting insight on resistance evolving over a three year period in two sites within Ouagadougou. A large number of bioassays has been conducted and specimens genotyped for different pyrethroid resistance mutation. Results allow to appreciate the speed of resistance evolving under strong selective pressure.

I would suggest the paper for publication and would recommend some minor corrections:

INTRODUCTION:

“The V1016I mutation confers resistance to pyrethroid insecticides only when associated with the

F1534C mutation”: Please specify that this is known for Ae. aegypti; for other species few data are available on this.

MATERIALS & METHODS

As stated by you most of the dosages you use were revisited by WHO recently; I would suggest to mention this in discussion since you may underestimate levels of resistance for some of your populations.

Why did you perform synergist assays using a different test protocol? This may be relevant also because you obtain very different results for pyrethroid bioassays at 1200Logements when using WHO-tube-test or CDC-bottle tests.

RESULTS

In the genotyping section and within the table referring to genotyping results please specify the number of specimens successfully genotyped. 

It is not clear whether you genotyped only specimens exposed to pyrethroids for pyrethroid resistance alleles or whether you genotyped all sampled specimens. 

This is relevant when interpreting results on the evolution of resistance alleles and the association of resistance alleles /genotypes/haplotypes with resistance phenotypes. It is obviously reasonable to include only genotyping results obtained for specimens exposed to pyrethroids when referring to the impact different resistance alleles /genotypes/haplotypes have on the possibility to survive pyrethroid exposure, but when referring to the evolution of resistance alleles /genotypes/haplotypes over the three year range it is definitely helpful to include as many specimens as possible ( and thus also those exposed to other insecticidal classes). 

“The frequency of the CIL/CIL triple mutant genotype increased from 2016 to 2018 (S1table) in 1200LG and Tabtenga, whereas the CVV/CVV genotype frequency decreased in both localities between 2016 and 2018 (S1table).”: please add frequencies in the text to avoid to look them necessarily up in S1.

DISCUSSION

Please rephrase the final sentence of the discussion section; it seems somehow contradictory since you already state before that LLIN probably have a high impact on selecting resistance .

Reviewer #2: In general, the manuscript is written in acceptable English but requires proofreading.

Minor Comment

The submitted manuscript does not include line numbering, which makes it difficult to make precise reference to comments in the text. Authors should take this into account in future submissions. 

In Figure S1 a and b of the 1200LG study site, the V/V genotypes are missing. The authors should correct this part.

On page 16, we suggest that authors write Long-Lasting Insecticidal Nets (LLINs) in full and not use an abbreviation the first time it is mentioned in the text. 

Major Comment

In this study, the authors mentioned in the results and discussion that PBO activity decreased in correlation with an increase in resistance over the 3 years. 

However, according to Figure 1 c and d, pre-exposure to PBO showed an increase in deltamethrin mortality from ~80% to 98% (1.22-fold) in 2016; from ~10% to 80% (8-fold) in 2017 and from ~5% to 70% (14-fold) in 2018 in the 1200 LG locality. In Tabtenga, pre-exposure to PBO also increased deltamethrin mortality from ~50% to 98% (1.96-fold) in 2016; from ~20% to 90% (4.5-fold) in 2017 and from ~3% to 70% (23-fold) in 2018. The same trends were observed with permethrin and PBO in both sites. 

Could the author provide a more detailed explanation for the decrease in oxidase activity mentioned in the manuscript over the three years?

Reviewer #3: PNTD-D-24-00505

review of Yameogo et al 2024:

Three years of insecticide resistance evolution and associated mechanisms in Aedes aegypti populations of Ouagadougou, Burkina Faso.

General comments:

In this paper, the authors evaluate the insecticide susceptibility of Ae. aegypti via metabolic and target site mechanisms over a 3-year surveillance period. For this they constituted 2 populations originated from 2 locations (Tabtenga and 1200LG). They performed bioassays to determine the susceptibility of the 2 populations to several insecticides. They also look for the presence ofseveral classic kdr mutation frequency. The results clearly showed a high resistance to the pyrethroid insecticides and the presence of kdr mutation that increased over the 3 years study. The results showed that insecticides from other family could be used as alternatives to the pyrethroids. 

In my opinion, the introduction should be improved and need some more information. For example, the authors mention vector control used against Aedes in the world and in Africa but are there any specific vector control interventions in Burkina Faso (adulticiding and / or larviciding / when)? Same, is it the first time that insecticide resistance levels and their mechanisms are measured/detected in Burkina Faso?

This study was carefully planned and conducted and the results are clearly presented in 2 figures and 3 tables. The paper is well written, the background of dengue epidemics by both Ae. aegypti and albopictus in central Africa is well documented as well the insecticide resistance mechanisms in these specific populations. I recommend this paper with minor revisions for publication in PNTD.

Minor comments:

Introduction:

More information on vector control in Burkina Faso are needed

Are there any previous studies on IR in BF?

The paragraphs regarding the kdr mutation should be regrouped in the introduction.

The metabolic resistance mechanism should be more detailed.

“Many reports…lacking” this sentence should be rephrased, it is not clear

Methods: 

It would be interesting to know to have an estimation of the number of larvae/pupae collected in the 2 localities to understand if the levels of resistance are coming from only few female mosquitoes or from more individuals. Is the number of individuals representatives of a large sampling area.

Why 2 different tests (WHO tube and CDC bottle) were used? The authors should explain this in the methods

Discussion:

The insecticide resistance levels were already high in 2016 indicating an intensive previous exposure to pyrethroids. The authors state that this could come from the treated Bednet distributions to target malaria vector. However Aedes aegypti is a day biting mosquito so the link between bednet presence and insecticide development in Aedes aegypti is not so obvious. Again vector control methods against Aedes mosquitoes but also against Anopheles and Culex should be more developed in the discussion. This would make the discussion clearer.

PLOS authors have the option to publish the peer review history of their article (what does this mean?). If published, this will include your full peer review and any attached files.

Reviewer #1: No

Reviewer #2: No

Reviewer #3: No

Figure Files:

Data Requirements:

Reproducibility:

References

---

## [Editor Report · Decision Letter 1]

5 Nov 2024

Dear dr Badolo,

We are pleased to inform you that your manuscript 'Three years of insecticide resistance evolution and associated mechanisms in Aedes aegypti populations of Ouagadougou, Burkina Faso.' has been provisionally accepted for publication in PLOS Neglected Tropical Diseases.

Best regards,

Jean-philippe David

Academic Editor

Nigel Beebe

Section Editor

Shaden Kamhawi

co-Editor-in-Chief

Paul Brindley

co-Editor-in-Chief

Dear Dr. Badolo,

First I apologize for the delay in handling this manuscript which was partially due to delayed reviewer's responses. In such context, I finally decided to personally handled the revised version of your manuscript.

After carefully reading the revised version of your manuscript, I found that most reviewers comments were adequaltely answered and that your manuscript in now suitable for publication in plos NTD.

Congratulations for this interesting work which provides useful information about insecticide resistance dynamics and assoicated mechanisms in Burkina Faso.

Best regards,

Jean-Philippe DAVID

---

## [Editor Report · Acceptance letter]

21 Nov 2024

Dear Prof Badolo,

We are delighted to inform you that your manuscript, "Three years of insecticide resistance evolution and associated mechanisms in <i>Aedes aegypti<i> populations of Ouagadougou, Burkina Faso.," has been formally accepted for publication in PLOS Neglected Tropical Diseases.

Best regards,

Shaden Kamhawi

co-Editor-in-Chief

Paul Brindley

co-Editor-in-Chief
